# Estimation of the Uncertainties Related to the Measurement of the Size and Quantities of Individual Silver Nanoparticles in Confectionery

**DOI:** 10.3390/ma12172677

**Published:** 2019-08-22

**Authors:** Nadia Waegeneers, Sandra De Vos, Eveline Verleysen, Ann Ruttens, Jan Mast

**Affiliations:** 1Service Trace Elements and Nanomaterials, Sciensano, Leuvensesteenweg 17, 3080 Tervuren, Belgium; 2Service Trace Elements and Nanomaterials, Sciensano, Groeselenberg 99, 1180 Brussels, Belgium

**Keywords:** single particle ICP-MS, silver nanoparticles, validation, complex matrices, measurement uncertainty

## Abstract

E174 (silver) is a food additive that may contain silver nanoparticles (AgNP). Validated methods are needed to size and quantify these particles in a regulatory context. However, no validations have yet been performed with food additives or real samples containing food additives requiring a sample preparation step prior to analysis. A single-particle inductively coupled plasma mass spectrometry (spICP-MS) method was developed and validated for sizing and quantifying the fraction of AgNP in E174 and in products containing E174, and associated uncertainties related to sample preparation, analysis and data interpretation were unraveled. The expanded measurement uncertainty for AgNP sizing was calculated to be 16% in E174-containing food products and increased up to 23% in E174 itself. The E174 food additives showed a large silver background concentration combined with a relatively low number of nanoparticles, making data interpretation more challenging than in the products. The standard uncertainties related to sample preparation, analysis, and challenging data interpretation were respectively 4.7%, 6.5%, and 6.0% for triplicate performances. For a single replicate sample, the uncertainty related to sample preparation increased to 6.8%. The expanded measurement uncertainty related to the concentration determination was 25–45% in these complex samples, without a clear distinction between additives and products. Overall, the validation parameters obtained for spICP-MS seem to be fit for the purpose of characterizing AgNP in E174 or E174-containing products.

## 1. Introduction

Humans are exposed to nanomaterials due to their wide application in many sectors, such as industry and food. A review by the European Food Safety Authority (EFSA) revealed 469 current and/or future food applications of engineered nanomaterials [1]. Food contact materials and food additives are the most reported current applications. Together with this exposure, questions arise about potential risks to consumers. Although they have a history of (safe) use, certain specific food additives have recently come under attention because they contain a fraction of particles in the nanosize range. Examples are synthetic amorphous silica (SAS), approved as food additive E551, titanium dioxide (E171) and silver (E174). 

Silver (Ag) is a food additive approved in the EU for use in the external coating of confectionery, for decoration of chocolates or in liqueurs [2]. Verleysen et al. [3] demonstrated that almost 95% of the particles eluted from an E174-containing confectionery product have a minimum external dimension smaller than 100 nm. Very few studies have estimated or measured exposure concentrations to the nanofraction present in the above-mentioned additives or food products containing them. This hampers the evaluation of potential risks related to the use of these food additives. In its recent re-evaluations, the EFSA ANS Panel recommended that specifications of the food additives laid down in Commission Regulation (EU) No 231/2012 should include characterization of the particle size distribution, as well as the percentage in number and mass of the particles in the nanoscale [4,5,6,7]. The EFSA guidance on risk assessment of the application of nanoscience and nanotechnologies in the feed and food chain recommends that prior to any risk assessment, a physicochemical characterization of potential nanomaterials—including thorough information on the particle size distribution—be carried out on the materials in their pristine state as well as when present in food products [8]. Data on particle size and size distribution should be provided as measured by at least two independent techniques, one being electron microscopy (EM), using standardized or validated methods. In the case of metal or metal oxide nanomaterials, single-particle inductively coupled plasma-mass spectrometry (spICP-MS) is often considered as a promising technique to characterize the elemental composition, size, and size distribution, at least for screening purposes [9,10,11,12,13]. By introducing a dispersion of metal or metal oxide (nano)particles into an ICP-MS system that is operated in time-resolved analysis mode, distinct ion plumes are generated in the plasma which are detected as signal pulses. The intensity of the pulses is related to the mass of the particles, while the number of pulses as a function of time is proportional to the particle number concentration. Hence, spICP-MS allows quantification of particle mass and number concentrations, and the particle size and size distribution can be estimated if additional information about particle shape and composition is known or assumed [11,14]. The method has recently been standardized for aqueous dispersions [15].

The detection and characterization of a pure nanomaterial in dispersion may be relatively straightforward, and is standardized for different techniques. However, the sample preparation of food products, bringing the (nano)particles into a dispersion prior to their characterization, is more challenging. Furthermore, the availability of (validated) analytical methods that are able to characterize nanomaterials in complex matrices is currently limited (EFSA, 2018). While several studies report on the validation of spICP-MS and transmission electron microscopy (TEM) analyses in simple dispersions (e.g., [3,9,13,16,17]), very few studies are available concerning the validation of spICP-MS or TEM analyses in complex food matrices. Peters et al. [18] validated spICP-MS for the sizing and quantification of silver nanoparticles in chicken meat, while Witzler et al. [19] validated the analysis of silver and gold nanoparticles in fruit juices. Dudkiewicz et al. [20] evaluated the measurement uncertainty of silver nanoparticles in chicken meat and of SAS in tomato soup. These validations in simple dispersions as well as in complex matrices have in common that they are performed with reference materials and/or monodisperse nanoparticle colloids spiked to the matrix. To our knowledge, no validations have yet been performed with food additives or real samples containing food additives that require a sample preparation step other than dilution prior to analysis. The objective of this study was therefore to estimate the additional variation (uncertainty) related to the sample preparation of samples with different complexity, to validate spICP-MS analyses in these complex sample types, and to estimate the measurement uncertainty for these analyses under routine operating conditions (i.e. a triplicate analysis performed on a single day). Silver (E174) was thereby selected as a case study. Knowledge about this additional variation caused by sample preparation will be beneficial to allow interpretation of existing validation data for spICP-MS as well as for other analytical techniques that rely of the same type of sample preparation. 

## 2. Materials and Methods 

### 2.1. Materials and Chemicals

The instrument parameters and operational conditions are described in Appendix A. Selectivity against matrix components (Appendix A) and other types of NPs, linearity (Appendix A), concentration (Appendix A), and the corresponding working range (Appendix A) were evaluated, and are described in Appendix A for single-particle ICP-MS. All TEM validation parameters are reported in Appendix A. 

Standards and chemicals—The reference material RM-8012 (gold nanoparticles with a nominal diameter of 30 nm) was obtained from the National Institute of Standards and Technology (NIST, Gaithersburg, MD, USA). The mean diameter reported on the certificate is 27.6 ± 2.1 nm (mean ± expanded uncertainty determined by TEM analysis(RM-8012, NIST, Gaithersburg, MD, USA)). Citrate-stabilized gold nanoparticles with a nominal diameter of 30-nm were obtained from nanoComposix (BiopureTM, San Diego, CA, USA). The mean diameter reported by the supplier is 31 ± 3 nm (determined by TEM analysis), with a coefficient of variation of 10.7%. The representative test material NM-300K (JRCNM03002a) is shown in Appendix A. It is an aqueous stock dispersion of noncoated silver nanoparticles, with stabilizing agents polyoxyethylene glycerol trioleate (4%) and Tween 20 (4%) and was obtained from the nanomaterial repository of the European Commission’s Joint Research Centre (JRC, Ispra, Italy). The silver content of the dispersion is reported to be 10.16% *w/w* in the material information sheet. As the material is very viscous, all preparations of dilutions were done by weighing to reduce pipetting errors. Single element TraceCERT® silver standard for ICP (994 ± 3 mg/L) and bovine serum albumin (lyophilized powder, ≥96%; BSA) were obtained from Sigma-Aldrich (Overijse, Belgium). Ethanol (EMSURE®) was obtained from Merck (Overijse, Belgium). Ultrapure water (UPW, 18.2 mΩ/cm) was prepared by using a Millipore Integral 3 system (Millipore, Molsheim, France) and used for sample preparation as well as dilution of standards and sample dispersions.

Samples—Three types of samples were examined in the validation study: the aqueous dispersion NM-300K, two silver food additives E174, and two food products containing the food additive E174 (confectionery and decorated chocolates), further denoted as “products”. The food additives and products were obtained from suppliers in Belgium, the United Kingdom, and Germany. The food additives consisted of 2-mm silver flakes (Ag-005) and 8-cm silver leaves (Ag-008). The food products were silver-coated chocolates (also known as mini “sugar beans”; Ag-P-003) and silver pearls containing mainly sugar and wheat flour/corn meal (Ag-P-002). All samples were stored in a clean, dry, and dark location. 

### 2.2. Sample Preparation for spICP-MS and TEM Analysis

The E174 food additives were dispersed according to a slightly modified version of the method of Jensen et al. [21]. In short, 0.05 g BSA was dissolved and diluted in 100 mL UPW to reach a final concentration of 0.5 mg/mL BSA. This solution was prepared freshly on each analysis day. An accurately weighed subsample of 0.0154 ± 0.0015 g of the E174 food additive was brought into a 20 mL liquid scintillation vial (Wheaton, Millville, NJ, USA). Subsequently, the sample was wetted dropwise with 30 µL ethanol (96%) and 0.970 mL of the fresh 0.5 mg/mL BSA solution. The mixture was gently swirled, after which another 5 mL of the BSA solution was added. The latter mixture was left to rest on ice for 5 min, after which it was sonicated for 16 min with a Vibracell VCX 750 sonicator (750 W, 20 kHz; Fisher Bioblock Scientific, Aalst, Belgium) with a 13 mm probe at 40% amplitude, which resulted in an applied energy of 32 ± 2 kJ (mean ± standard deviation; preparation for spICP-MS analysis) or 37 ± 1 kJ (preparation for TEM). Three independent replicates were prepared per food additive sample for spICP-MS analysis, while one replicate was prepared for TEM analysis. 

The products were prepared in a similar way for spICP-MS analysis: three items of the silver-coated chocolates (about 2 g) or 12 items of the silver pearls (about 2 g) were accurately weighed and brought into a 50 mL polypropylene vial (Sarstedt, Nümbrecht, Germany). The samples were wetted dropwise with 125 µL ethanol (96%), after which 20 mL (chocolates) or 25 mL (pearls) of the freshly prepared BSA 0.5 mg/mL solution was added. The samples were shaken in a Multi Reax (Heidolph Instruments, Schwabach, Germany) until either the silver layer was completely removed from the chocolate core (silver-coated chocolates), or the whole product was dissolved (pearls). In the case of the silver-coated chocolate, the remaining cores were rinsed with another 5 mL of the BSA 0.5 mg/mL solution, after which the cores were removed from the dispersion. The weight of the removed cores was noted as well. The dispersions were further processed as the E174 food additive dispersions. Three independent replicates were prepared per food product sample. For TEM analysis, the products were prepared as specified in [3] in one replicate.

The NM-300K dispersion did not require any sample preparation for spICP-MS. Three replicates were sampled directly from the dispersion after vortex stirring, as described below. For TEM analysis, NM-300K was prepared as specified in [3] in one replicate.

All dispersions were vortex stirred for 30 s prior to dilution for spICP-MS analysis. Each dispersion was diluted in polypropylene vials to two different levels with UPW. The two appropriate dilutions were determined after a range finding test, during which different dilutions of the dispersion were measured. The two dilutions had to result in a proportionally changing number of detected particles, a constant particle size, and 200–2200 detected particles. All vials and polypropylene tubes were washed with acid prior to use.

### 2.3. Instrumentation and Analysis

ICP-MS/MS (Agilent 8800, Agilent Technologies, Santa Clare, CA, USA) was used for data acquisition in time-resolved analysis mode. Silver was measured at *m/z* 107. In order to increase the sensitivity of the spICP-MS analyses, instrument tuning was optimized for 107Ag by adjusting, amongst others, sample depth and carrier gas flow rate. The instrument parameters and operational conditions are described in Appendix A. Specific instrument settings applied for spICP-MS are given in Appendix A. The transport efficiency was determined daily according to the particle frequency method [14] using the reference material NIST-8012 (gold nanoparticles, nominal size 30 nm) or 30 nm-gold nanoparticles from nanoComposix, both at a concentration of 12.5 ng/L under the same instrumental conditions as the samples. Mass calibration was performed by measuring ionic Ag standard solutions. Following the measurement of each sample and extensive rinsing, UPW was analyzed to verify that no carry-over of Ag from the previous sample occurred. 

The single particle calculation spreadsheet described by Peters et al. [13] was used to calculate particle size distributions and particle number (C_p_) or mass concentrations (C_m_). The particle diameter, referred to as equivalent spherical diameter (ESD), was obtained from the particle mass assuming a spherical geometry. A detailed description of the single particle calculation spreadsheet, including calculation equations, can be found in [13]. To discriminate between particles and ionic Ag or incomplete particle events (ion plumes detected over two consecutive dwell times), an iterative algorithm based on *µ + n × σ* was applied on the data as described by Mitrano et al. [12] and Tuoriniemi et al. [22], whereby *µ* and *σ* are, respectively, the mean measured value and its standard deviation, and *n* is an integer varying between 4 and 5 depending on the sample.

TEM specimens were prepared using Alcian-blue-treated positively charged pioloform- and carbon-coated 400 mesh copper grids (Agar Scientific, Essex, England), based on the SOP “Preparation of EM-grids containing a representative sample of a dispersed NM” [23] for E174 food additives or as described in [3] for products. TEM imaging was done using a Tecnai G2 Spirit TEM with BioTwin lens configuration (FEI, Eindhoven, The Netherlands), based on the SOP “Transmission electron microscopic imaging of nanomaterials” which aims to record a set of calibrated transmission electron images that representatively show the NM on the TEM specimen [23]. Images were recorded with a 4 × 4 k Eagle CCD camera (FEI, Eindhoven, the Netherlands) using the TEM imaging & analysis (TIA) software (FEI, Eindhoven, The Netherlands).

For each material, a suitable magnification allowing the measurement of a sufficiently high number of particles for descriptive and quantitative image analyses was selected. 

Image analysis was done based on the image analysis SOP “Measurement of the minimum external dimension of the primary particles of particulate materials from TEM images by the NanoDefine ParticleSizer software” [24,25]. NM-300K was analyzed in default mode, and pristine E174 materials and products were analyzed based on a combination of image analysis modes. The approach combining the SOPs for TEM specimen preparation, TEM imaging, and image analysis was evaluated on a series of certified reference materials and representative test materials, including NM-300K (See Appendix A), with varying physical properties, including particle size, shape, and agglomeration state by Verleysen et al. [26]. Silver particles measured by TEM analysis were identified based on their mass–thickness contrast and morphology. The image analysis settings were optimized for the analysis of silver NPs taking in account the robustness against small variations in the image analysis settings, as illustrated in Appendix A for the optimization of the local threshold (Appendix A). The chemical composition of the measured particles was verified using scanning TEM (STEM) combined with energy dispersive X-ray spectrometry (EDX) analysis on representative samples [27]. Large micrometer-sized silver flakes of E174 were not included in the size measurement since they are much larger than the upper limit of quantification of the EM and spICP-MS analyses. This had a negligible effect on the estimation of the (median) size since the number of such flakes was low compared to the number of (nano)particles.

### 2.4. Determination of Performance Characteristics

Precision—Repeatability and intermediate precision (within-lab reproducibility) were determined using the following setup: on each of five different days, three independent replicates of NM-300K, two E174 food additives (Ag-005 and Ag-008), and two food products containing E174 (Ag-P-002 and Ag-P-003) were prepared as described in Section 2.2 and subsequently diluted for spICP-MS measurement. 

Repeatability and intermediate precision were assessed via one-way analysis of variance. Repeatability standard deviation (s_r_), between-day standard deviation (s_d_), and intermediate precision standard deviation (s_ip_) were calculated according to the equations below:
(1)$$s_{r}=\;\sqrt{\mathrm{MSW}},$$
(2)$$s_{d}=\;\sqrt{\frac{\mathrm{MSB}-\mathrm{MSW}}{n}},$$
(3)$$s_{\mathrm{ip}}=\;\sqrt{s_{r}^{2}+{\;s}_{d}^{2}}.$$
MSW is the mean squares within days, MSB is the mean squares between days, and n is the number of measurements per day (three replicates).

Trueness/recovery—Single-particle ICP-MS is considered to be a screening tool in nanoparticle analysis because of (1) its assumptions about the shape and composition of the nanoparticles, and (2) its relatively high size detection limits which might result in a substantial amount of nanoparticles not being detected. Single-particle ICP-MS results can be complemented with TEM results to better estimate the number-based size distribution. In screening methods, the determination of trueness is not a necessary validation parameter. Nevertheless, an attempt was made to quantify the bias and the uncertainty on the bias. Since certified reference materials with size distribution and concentration values certified for spICP-MS are not yet available, quantification of the bias remains dependent on proxy methods. As a proxy for trueness estimation (“apparent trueness”), the median particle size (ESD) and ESD size distribution determined by spICP-MS were compared to median size and size distribution determined by TEM for NM-300K. Given that in a regulatory context the minimum external dimension has to be estimated, the minimum Feret diameter (Feret-min; the minimum distance of parallel tangents at opposing particle borders) was thereby preferred as a TEM size parameter over the equivalent circular diameter (ECD; the diameter of a circle that has the same area as the projection area of the particle). The recovery of particle mass and particle number concentrations were determined as a percentage of the theoretical concentrations, which were estimated as follows. The reported total silver content of NM-300K is 10.16% (*w/w*). Preliminary experiments with a specific silver electrode revealed that in a 500-fold diluted NM-300K dispersion, 3–4% of the total Ag was present in ionic form, while in a million-fold dilution 22% of the total Ag was ionized [28]. Additionally, a fraction of the nanoparticles in the NM-300K dispersion is too small to be detected by spICP-MS. This fraction represents on average 14% in number (i.e., the percentile of Feret-min values <12 nm in the Feret-min size distribution) and, given the small size of these particles, 1.5% in mass. Assuming all particles are spherical with a diameter of 15.4 nm (Table 1) and a density of 10.49 g/cm^3^, the mass of a silver particle is 2.0 × 10^−17^ g. Taking into account the above considerations about ionization, detection, and particle shape, the mass and number concentration in the NM-300K dispersion was estimated as 79 mg/g and 3.4 × 10^15^ particles/g, respectively.

For particle size (median ESD), the uncertainty on the bias was estimated as:(4)$$u_{\Delta }=\;\sqrt{{\left(\frac{s_{t}}{\sqrt{n_{t}}}\right)}^{2}+{\left(u_{\mathrm{TEM}}\right)}^{2}},$$
where s_t_ is the standard deviation of the results of the apparent trueness determination experiments, n_t_ is the number of measurements in the trueness determination experiments (n_t_ = 15), and u_TEM_ is the combined uncertainty of the Feret-min determination by TEM.

For particle number and mass concentration, the relative uncertainty on the bias was estimated as:(5)$$u_{\Delta }=\;\frac{s_{t}}{\sqrt{n_{t}}}.$$

To determine the measurement uncertainty under routine measurement conditions, the repeatability and between-day variations were divided respectively by the number of replicates and the number of measurement days that are applied under routine conditions. Hence, the combined measurement uncertainty (u_c_) was calculated as: (6)$$u_{c}=\;\sqrt{\frac{s_{r}^{2}}{n}+\;\frac{s_{d}^{2}}{d}+u_{\Delta }^{2},}$$
where d is the number of days and n is the number of replicates per day. The expanded measurement uncertainty (U) was calculated as: (7)$$U=k\cdot u_{c},\;\mathrm{with}\;k\;=\;2.$$

Relative uncertainties were calculated by dividing the calculated uncertainties by the mean ESD, number, or mass concentration, respectively.

Selectivity against matrix components (Appendix A), robustness of the analytical procedure (Appendix A), linearity (Appendix A), limits of detection and quantification of particle size (Appendix A) The selection of the parameters measured by TEM analysis is justified by cross correlation analysis (Appendix A). The uncertainty contributions to the combined and expanded measurement uncertainties for TEM measurement of the median Feret-min, ECD, aspect ratio and solidity are summarized in Appendix A.

## 3. Results

### 3.1. Size Determination

The median ESD of silver nanoparticles (AgNP) in the examined samples varied between 15 and 27 nm (Table 1). The expanded measurement uncertainty related to the size determination of dispersed AgNP by spICP-MS under routine conditions (i.e., triplicate analysis on a single day) was calculated to be 13% (NM-300K; Table 1). This is similar to the size determination by TEM, despite the use of more replicates (three versus one; Appendix A). The expanded measurement uncertainty increased to 16% in the products and up to 23% in pristine E174 (Table 1). 

Aqueous dispersions of nanoparticles such as NM-300K do not require any sample preparation prior to spICP-MS analysis. Hence, the observed uncertainty can be attributed to the spICP-MS measurement, including the necessary sample dilution prior to measurement and calculations. The sample preparation of complex matrices increased the uncertainty, as shown here for the products. Although the sample preparation was similar for pristine E174 as for the products containing the additive, the associated uncertainty was larger for pristine E174 than for the products. SpICP-MS was more challenging for the E174 additives than for the products because of the relatively low number of detected particles and the high background signal—factors that made particle detection and counting statistics less reliable. Combined with a larger sample heterogeneity (see below), this can explain the larger uncertainty.

The repeatability standard deviation (s_r_) of the ESD determination increased from 1% for NM-300K to 4–6% for most of the complex matrices. This indicates that sample preparation was less repeatable than sample dilution. The high s_r_ of 11% for Ag-005 was due to sample inhomogeneity itself, as demonstrated by confirmatory analyses. Each of the three replicates was in turn diluted in triplicate, and each dilution was measured twice by spICP-MS to differentiate the variation introduced by the measurement, the dilution before measurement, and the sample itself (see Appendix A for detailed results). The repeated analyses of the same dilution and the repeated dilutions of the same replicate all gave similar results. Only between the three independently prepared replicates (under repeatability conditions) did the results differ.

The between-day standard deviation (s_d_) increased from 1.5% (products) to 8% (additives). Neither sample preparation nor sample dilution seemed to be the driving factors in between-day variation. Between-day variation may also have been caused by differences in data interpretation of the ICP-MS signals, particularly in relation to the determination of the limit for particle detection, which is the discrimination between particle signals and the background signal. Data interpretation is difficult in complex samples where the particle size distribution overlaps with the background signal. Especially for the E174 additives, it was challenging to determine the optimal dilution and to determine the limit for particle detection. Due to the high background signal and low number of detected particles, there was not always a proportional relation between the results of both applied dilution levels. In addition, the determination of the limit for particle detection was not automated during our data analyses. On the one hand, this may have induced slight differences in data interpretation between days; on the other hand, one single experienced person interpreted all data, potentially leading to an underestimation of the intermediate precision. Automation of the limit of particle detection determination may help to reduce variation between days, although in our experience no single algorithm works for all types of nanoparticle–matrix combinations, or sometimes even within a nanoparticle–matrix combination. Hence, verification by an experienced person remains necessary. 

Characterization of the Ag nanoparticles by TEM allowed the confirmation of the assumptions made in spICP-MS analysis about the spherical shape of the particles. The aspect ratio was 1.08 for NM-300K, while for E174 and the products the aspect ratio varied between 1.07 and 1.28, which confirms that the AgNP were near spherical and justifies the use of the ESD as estimator of the minimum external dimension of the AgNP in a regulatory context. Because the size quantification limit for the AgNP analyses varied between 11 and 23 nm among the different sample types and analysis days, a fraction of the small sized particles measured in TEM analyses was not detected by spICP-MS. This fraction was calculated based on TEM results of the same samples, allowing a total number recovery correction when calculating normalized number-based size distributions for spICP-MS. The dispersed NM-300K particles followed a normal number-based size distribution, both for the ESD determined by spICP-MS and for the Feret-min determined by TEM (Figure 1a). Despite the missing nanoparticle fraction, the ESD size distribution showed a high degree of overlap with the Feret-min size distribution, with distribution modes and half width at half maximum values for ESD being 15.4 ± 0.4 nm and 2.6 ± 0.2 nm and for Feret-min 15.0 ± 0.2 nm and 2.9 ± 0.1 nm (mean over five validation days ± standard deviation). The high degree of overlap between both distributions suggests a lack of bias in the ESD size distribution and justifies the use of the median Feret-min (15.4 nm; Table 1) as reference value to estimate the apparent bias on the median ESD. This apparent bias was −0.1 nm (or −0.8%) for NM-300K. The uncertainty on the bias estimation was 0.9 nm, or 6.0% (Table 1). The latter value was used in the calculation of the measurement uncertainty on the ESD determination of AgNP in E174 and in products.

The particles present in the E174 samples and products showed a log-normal distribution with a tailing on the right-hand side (Figure 1b,c and Figure A1a,b). Again, a high degree of overlap was seen between the ESD and the Feret-min distributions, although the overlap was better for the products than for the additives. 

### 3.2. Particle and Mass Concentration

The particle concentration (mass and number based) varied over four to five orders of magnitude among the different samples (Table 2 and Table 3). The expanded measurement uncertainty related to the concentration determination of dispersed AgNP (NM-300K) by spICP-MS was about 20% under routine conditions (triplicate analysis under repeatability conditions) and similar for the number and mass concentration (Table 2 and Table 3). The uncertainty increased to 25–45% in complex samples, without a clear distinction between additives and products. 

The repeatability of the particle concentration data among all the complex sample types was rather similar (16–29%) and at least a factor of two larger than the repeatability in the colloidal dispersion. The repeatability variation was also larger than the between-day variation in the complex samples, which ranged between 6% and 20%. In Appendix A, analysis of the variation sources among replicate analyses of Ag-005 (E174, 2 mm flakes) allowed differentiating the sources of repeatability variation. Appendix A illustrates that sampling and sample preparation contributes more to repeatability variation than dilution and measurement All this may reflect the heterogeneity of these complex samples, but as the repeatability variation was also much larger for the concentration analysis than for the size determination, it suggests that the sample preparation protocol is optimal to obtain a dispersion that is homogenous in terms of size distribution, but sub-optimal to obtain a dispersion that is homogenous in terms of concentration sampling.

A general problem in the trueness assessment of silver nanoparticle concentration analysis is the lack of availability of suitable certified reference materials or other techniques to determine the nanoparticle concentration. An attempt was made to calculate the recovery as a percentage of the theoretical mass and number concentrations. The measured mean Ag particle mass and number concentrations were respectively 44 mg/g and 1.73 × 10^15^ particles/g of NM-300K dispersion. This corresponds to an apparent recovery of 57% for the mass concentration and 51% for the number concentration. However, given the uncertainty around the dissolution of Ag in diluted NM-300K dispersions [28], these apparent recoveries were not considered as reliable estimates of the bias with regard to particle mass and number concentration.

## 4. Discussion

The validation of the analysis of AgNP in E174 and confectionery, which are complex matrices, by spICP-MS followed a top-down approach. That is, the measurement uncertainty was derived from observed variations during the whole methodological procedure. This methodological procedure can be segmented according to different stages, each having its own error, ordered as follows:

sampling → dispersion preparation → dilution → measurement → data interpretation.

The combined uncertainty related to the spICP-MS analysis of NM-300K covers the dilution, measurement, and data interpretation stages (i.e., the analysis component), while the uncertainty related to the analysis of products covers the same stages, in addition to the sampling and dispersion preparation uncertainties (i.e., the sample preparation component). Hence, the latter measurement uncertainty can be segmented into following components:(8)$$u_{c}=\;\sqrt{u_{\mathrm{sample}\;\mathrm{preparation}}^{2}+{\;u}_{\mathrm{analysis}}^{2}}.$$

From Equation (8) and the uncertainties related to the size determination of NM-300K (13%/2 = 6.5%) and products (16%/2 = 8%; Table 1), it can be derived that the standard uncertainty related to the sample preparation of these complex samples would be 4.7% (k = 1) if the sample preparation were performed in triplicate. This increases to 6.8% (k = 1) if only one replicate were to be used (for detailed calculations and data, see Appendix A). The calculated contribution of the sample preparation to the measurement uncertainty of the size determination can be applied to other existing spICP-MS validations where sample preparation was not taken into account to get a better estimate of the uncertainties related to the sizing of AgNP in product samples. It can be questioned whether the obtained uncertainties also apply to matrices other than E174 and confectionery products. Nevertheless, they are valid for all food matrices in which silver is currently allowed in the European Union [2], and are probably also valid for gold nanoparticles in E175, which is also applied in confectionery. The calculated contribution of the sample preparation to the measurement uncertainty of the size determination can also be applied to other techniques that use a similar sample preparation protocol to determine the size of AgNP in complex matrices, using an alternative method such as TEM [27]. If the uncertainty in sample preparation were applied to the TEM data in Table 1, the expanded measurement uncertainty on the Feret-min of AgNP in E174 and products would be on the order of 18%. This corresponds to the lower end of uncertainties derived by Dudkiewicz et al. (2015), who calculated expanded measurement uncertainties of 20% and 44% in chicken paste spiked with AgNP suspensions. Although these uncertainties expressed as percentage seem high, they are only a few nanometers in absolute terms. Results with uncertainties are essential for product characterization, in a regulatory context and in risk analysis.

This approach assumes that the measurement uncertainties of the size measurands—ESD for spICP-MS and Feret-min for TEM—are similar. Depending on the shape of the particles, their values can be different. TEM analyses of the examined samples showed median aspect ratios ranging from 1.07 to 1.21 for solidities ranging from to 0.97 to 0.99 (38), suggesting near-spherical morphology for the majority of particles. For near-spherical particles, TEM validation studies showed that measurement uncertainties of Feret-min and ECD, a proxy for ESD, were very similar (Appendix A, [16,26]) confirming this assumption. For particles with aberrant shape, a strategy considering the associated additional uncertainty is worth investigating.

Validation performance characteristics with regard to the analysis of AgNP by spICP-MS are scarce in the literature, and to this point had been derived only by spiking standard particles in water, fruit juices, or chicken meat [13,18,19]. In these studies, the repeatability and intermediate precision standard deviations in water ranged respectively from 0.3% to 2.3% and from 2% to 5% for sizing, and showed little dependence on the size of the nanoparticles. The repeatability and intermediate precision standard deviation for sizing NM-300K in the current study fit well within these ranges (Table 1). 

The applied sample preparation was based on the protocol of Jensen et al. [21], and is a well-established sample preparation protocol for nanotoxicological testing. It is a compromise between very mild and harsh sample preparation, and cannot exclude the adsorption of particles to food compounds and to larger Ag particles. The proposed measurement uncertainties associated with the applied sample preparation cannot be extrapolated directly to mild sample preparation (e.g., elution with water only [3] with possible extensive adsorption), nor to harsh sample preparation protocols, which can induce the de novo formation of particles or particle breakdown due to the relative physical and chemical instability of silver [29].

Repeatability and intermediate precision in complex matrices seem to be more size (distribution) dependent. The repeatability of sizing 60 nm AgNP in chicken meat and 50 nm AgNP in fruit juices varied between 0.8% and 2.6%, while it varied between 1.5% and 8% for 30 nm AgNP in fruit juices [18,19]. Similarly, intermediate precision for the larger AgNP in different matrices varied between 2.2% and 6.6%, while it varied between 10% and 25% for the 30-nm AgNP in fruit juices. Sample preparation, including sampling, of chicken meat thereby did not seem to have any influence on the repeatability, and had only a small influence on the intermediate precision of the size determination by spICP-MS [18]. Our results demonstrated a clear contribution of sampling and sample preparation to the size determination of 20–30 nm AgNP in additives and products. This can be partially attributed to the smaller size of the detected particles in these complex samples, but it is more likely that the larger heterogeneity of real samples compared to spiked samples is responsible for the discrepancy. 

The precision of the concentration analysis by spICP-MS seems to be similar whether it is mass-or number-based, as demonstrated in spiked chicken meat [18] and in the current study. Any dependency of the concentration precision on the size of the particles is less pronounced [19]. Taking our results into account, which are situated between those of the chicken meat and fruit juice validation, it seems that the precision of spICP-MS concentration analyses is mainly related to the type of matrix. 

Trueness can only be validated by means of certified reference materials, which are not yet available for spICP-MS analyses. As an alternative, previous validation studies made comparisons with the nominal median diameter and/or concentration of the spiking standards. For real samples, no comparison with nominal values can be made. Therefore, we compared the size distributions obtained by two different techniques, and this proved that the size distributions of near-spherical AgNP were similar between spICP-MS and TEM. In the absence of certified reference materials, this is a recommendable methodology in the case of unknown complex samples. The product samples were sonicated prior to spICP-MS, but not prior to TEM analysis. This hardly influenced the results and conclusions, as experiments at different sonication energies have demonstrated that there was only a minor difference in the Feret-min values whether sonication had been applied during sample preparation or not [27]. 

Overall, the values of the validation parameters obtained for spICP-MS analyses in real samples can be related to the larger complexity of analyzing real samples compared to spiked standards, and seem to be fit for our purpose. Single-particle ICP-MS is often viewed as a promising technique to characterize metallic nanoparticles [9,10,11,12,13], and the current study corroborates this point of view. Nevertheless, spICP-MS can also be challenging in the case of unknown samples, especially when it is operated at its limits. The measurement uncertainty related to the analysis of E174 calculated in the present study covers data interpretation at its limits, as well as sample preparation and analysis. By adding a term representing data interpretation uncertainty to Equation (8), it can be calculated that the standard uncertainty related to the difficulty of data interpretation in the case of high background/low particle numbers amounted to 6.0% (detailed calculation see Appendix A). As mentioned earlier, deciding on the limit of particle detection in the case of overlapping distributions between background and particle signal is a non-negligible source of variation. 

## 5. Conclusions

The sizing and quantification of AgNP were validated for the first time for native particles present in confectionery samples. Single-particle ICP-MS was used to unravel the variation introduced by sample preparation, analysis, and data interpretation of E174 food additives and products containing E174. The validation parameters obtained for spICP-MS seemed to be fit for our purpose, with expanded measurement uncertainties (k = 2) for AgNP sizing of 16% in E174-containing food products and up to 23% in the food additive itself. The uncertainty associated with sample preparation was 4.7% (k = 1) in the case of triplicate analyses under repeatability conditions and 6.8% in the case of a single replicate analysis. This variation caused by sample preparation is also suitable for use with other analytical techniques that rely on the same type of sample preparation, such as TEM. The expanded measurement uncertainty related to the concentration determination of AgNP was 25–45% in complex samples, without a clear distinction between additives and products. Trueness validation in these complex samples remains a problem—especially for the concentration analysis. There remains an urgent need for certified reference materials suitable for spICP-MS. The validation results will be applied to a study of E174 food additives and E174-containing products that are available on the market, in which both TEM and spICP-MS will be used to characterize the AgNP. 

## Figures and Tables

**Figure 1 materials-12-02677-f001:**
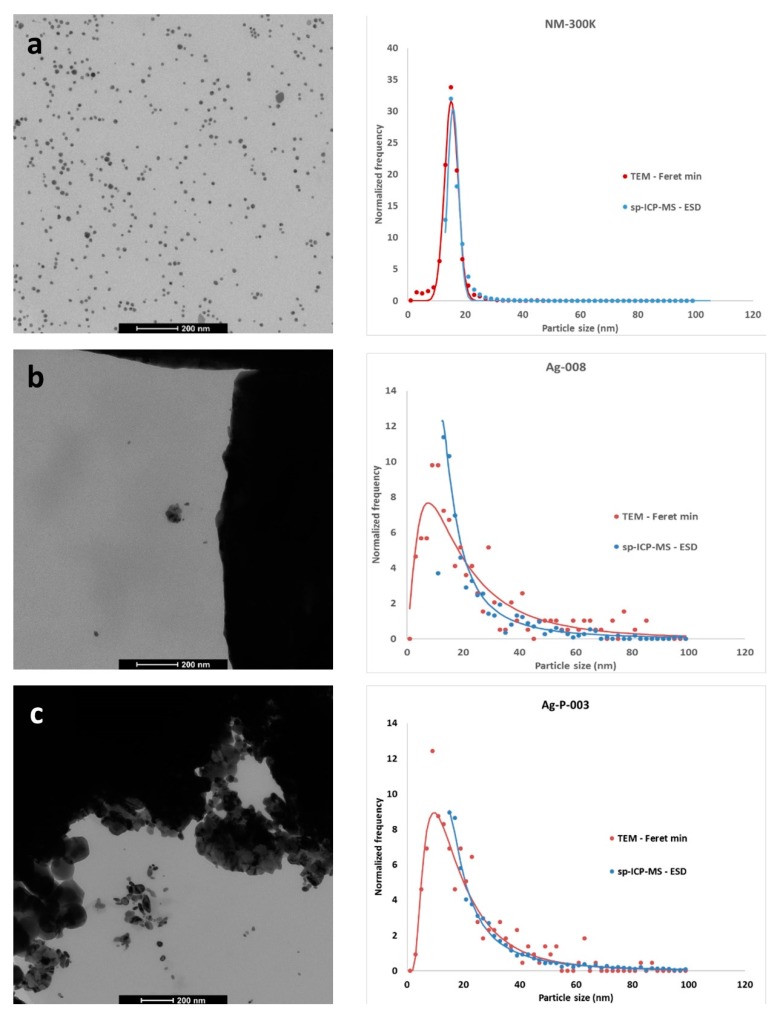
Representative TEM micrographs and silver particle size distributions determined by TEM and single-particle inductively coupled plasma-mass spectrometry (spICP-MS) for (**a**) a colloidal dispersion of silver nanoparticles (AgNP) (NM-300K); (**b**) a pristine E174 food additive (Ag-008); and (**c**) an E174-containing food product (Ag-P-003). ESD: equivalent spherical diameter.

**Table 1 materials-12-02677-t001:** Mean of the median nanoparticle sizes and their associated standard deviation, repeatability (s_r_), between-day variation (s_d_), and intermediate precision (s_ip_). Contributions of the trueness uncertainty (u_t_), within-day (u_r_), and between-day (u_d_) uncertainty to the expanded uncertainty (U) on the measurement of the median size by TEM (Feret-min) and spICP-MS (ESD) under routine measurement conditions (1 replicate × 1 day for Feret-min and 3 replicates × 1 day for ESD).

									Routine Conditions		
Sample	Sample Code	Size Parameter	Mean of Median Size (nm)	Stdev (nm)	s_r_ (%)	s_d_ (%)	s_ip_ (%)	u_t_ (%)	u_r_ (%)	u_d_ (%)	U (k = 2; %)
Aqueous dispersion of AgNP	NM-300K	Feret-min	15.4	0.5	2.4	2.5	3.4	4.8	2.4	2.5	12
		ESD	15.3	0.4	0.9	2.8	3.0	6.0	0.5	2.8	13
E174 2-mm flakes	Ag-005	ESD	26.7	1.3	11	7.3	13	6.0	6.3	7.3	23
E174 8-cm leaves	Ag-008	ESD	18.7	1.5	4.2	7.8	8.9	6.0	2.4	7.8	20
Silver pearls containing E174	Ag-P-002	ESD	19.4	0.7	5.5	1.5	5.7	6.0	3.2	1.5	14
Silver-coated chocolates containing E174	Ag-P-003	ESD	22.5	1.2	6.2	3.4	7.1	6.0	3.6	3.4	16

**Table 2 materials-12-02677-t002:** Mean particle number concentrations (C_p_) and their associated standard deviation, repeatability (s_r_), between-day variation (s_d_), and intermediate precision (s_ip_). Contributions of the trueness uncertainty (u_t_), within-day (u_r_), and between-day (u_d_) uncertainty to the expanded uncertainty (U) on the measurement of the particle number concentration by spICP-MS under routine measurement conditions (3 replicates × 1 day).

								Routine Conditions		
Sample	Sample Code	Mean C_p_ (kg^−1^)	Stdev (kg^−1^)	s_r_ (%)	s_d_ (%)	s_ip_ (%)	u_t_ (%)	u_r_ (%)	u_d_ (%)	U (k = 2; %)
Aqueous dispersion of AgNP	NM-300K	1.7 × 10^18^	0.2 × 10^18^	7.8	8.9	12	3.3	4.5	8.9	21
E174 2-mm flakes	Ag-005	6.9 × 10^15^	0.8 × 10^15^	22	13	25	3.3	13	13	36
E174 8-cm leaves	Ag-008	1.2 × 10^16^	0.06 × 10^16^	17	6.6	18	3.3	9.9	6.6	25
Silver pearls containing E174	Ag-P-002	4.4 × 10^13^	1.1 × 10^13^	16	12	20	3.3	9.2	12	31
Silver-coated chocolates containing E174	Ag-P-003	4.2 × 10^13^	1.1 × 10^13^	19	20	27	3.3	11	20	45

**Table 3 materials-12-02677-t003:** Mean particle mass concentrations (C_m_) and their associated standard deviation, repeatability (s_r_), between-day variation (s_d_) and intermediate precision (s_ip_). Contributions of the trueness uncertainty (u_t_), within-day (u_r_), and between-day (u_d_) uncertainty to the expanded uncertainty (U) on the measurement of the particle mass concentration by spICP-MS under routine measurement conditions (3 replicates × 1 day).

								Routine Conditions		
Sample	Sample Code	Mean C_m_ (g/kg)	Stdev (g/kg)	s_r_ (%)	s_d_ (%)	s_ip_ (%)	u_t_ (%)	u_r_ (%)	u_d_ (%)	U (k = 2; %)
Aqueous dispersion of AgNP	NM-300K	44	4	9.6	6.4	12	2.9	5.6	6.4	18
E174 2-mm flakes	Ag-005	2.3	0.2	29	11	31	2.9	17	11	41
E174 8-cm leaves	Ag-008	1.9	0.2	19	6.3	20	2.9	11	6.3	26
Silver pearls containing E174	Ag-P-002	0.008	0.001	19	13	23	2.9	11	13	34
Silver-coated chocolates containing E174	Ag-P-003	0.010	0.001	23	9.0	25	2.9	14	9.0	33

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
