# Peer review of "Estimation of the Uncertainties Related to the Measurement of the Size and Quantities of Individual Silver Nanoparticles in Confectionery"

_materials, 2019, doi:10.3390/ma12172677_

Round 1
Reviewer 1 Report
In this article, the uncertainties of the size and quantities of silver nanoparticles in confectionery samples. Different sources of the variations from sample preparation, analysis, and data interpretation are investigated on the E174 additives and products. This is a useful work to characterize variations, which could be potentially published on Materials. However, serval comments need to be addressed.
You should define the abbreviations before using them, such as ICP-MS, and spICP-Ms in the Abstract.
As shown in the TEM images, the particle sizes are irregular, the size of a single particle is different, which is depending on how you characterize the size of the particle. For example, you may get a smaller size if it is measured along the short dimension of the particle. Please explain how the size of the nanoparticle is determined and add the discussion into the manuscript. An extra variation may be introduced when using different methods to determine the size of particles.
When using the spICP-MS method, there is a size quantification limit for AgNPs in the range from 11 nm to 23 nm, depending on the sample types. What is the potential way to reduce the limit or the reasonable alternative method to get the information of AgNPs which could not be detected by spICP-MS?
Author Response
Reviewer 1:
You should define the abbreviations before using them, such as ICP-MS, and spICP-Ms in the Abstract.
The abbreviation was were defined before using them (line 13).
When using the spICP-MS method, there is a size quantification limit for AgNPs in the range from 11 nm to 23 nm, depending on the sample types. What is the potential way to reduce the limit or the reasonable alternative method to get the information of AgNPs which could not be detected by spICP-MS?
The fraction of the small sized particles can be measured by TEM. This is described in lines 295-299.
“Because the size quantification limit for the AgNP analyses varied between 11 and 23 nm among the different sample types and analysis days, a fraction of the small sized particles, measured in TEM analyses, was not detected by spICP-MS. This fraction was calculated based on TEM results of the same samples, allowing a total number recovery correction when calculating normalized number-based size distributions for spICP-MS.”
As shown in the TEM images, the particle sizes are irregular, the size of a single particle is different, which is depending on how you characterize the size of the particle. For example, you may get a smaller size if it is measured along the short dimension of the particle. Please explain how the size of the nanoparticle is determined and add the discussion into the manuscript. An extra variation may be introduced when using different methods to determine the size of particles.
The measurands are explained in the “Materials and methods” section, lines 210 – 216. To address the remark of the reviewer, following sentences are introduced in the discussion, after line 395:“This approach assumes that the measurement uncertainties of the size measurands, ESD for spICP-MS and Feret Min for TEM, are similar. Depending on the shape of the particles, their values can be different. TEM analyses of the examined samples showed median aspect ratios ranging from 1.07 to 1.21 for solidities ranging from to 0.97 to 0.99 (38), suggesting near-spherical morphology for the majority of particles. For near-spherical particles, TEM validation studies show that measurement uncertainties of Feret Min and ECD, a proxy for ESD, are very similar (Table S6, [16], [26]) confirming this assumption. For particles with aberrant shape, a strategy taking in account associated additional uncertainty is worth investigating.”

Reviewer 2 Report
This manuscript discussed the sizing and quantification of AgNP in confectionery using SP-ICP-MS. The results are interesting and worth publishing. However, the reviewer has one concern about the method applied in this study. The efficiency of AgNP extraction from food additive and products was not investigated in this study. Especially for the products, the authors did not consider the AgNPs sorbed by the food. This may significantly affect the sizing and quantification of AgNPs. An effective way to know the efficiency is to totally digest samples using strong acid and quantify total Ag content in the samples. Then the authors could know the extraction efficiency by calculating the mass ratio of AgNPs to total Ag.
Author Response
Reviewer 2
This manuscript discussed the sizing and quantification of AgNP in confectionery using SP-ICP-MS. The results are interesting and worth publishing.
However, the reviewer has one concern about the method applied in this study. The efficiency of AgNP extraction from food additive and products was not investigated in this study. Especially for the products, the authors did not consider the AgNPs sorbed by the food. This may significantly affect the sizing and quantification of AgNPs. An effective way to know the efficiency is to totally digest samples using strong acid and quantify total Ag content in the samples. Then the authors could know the extraction efficiency by calculating the mass ratio of AgNPs to total Ag.
We agree with the reviewer that adsorption of particles to food compounds and to larger Ag particles cannot be excluded and is a concern.
Alternative sample preparation methods aiming to liberate possibly sorbed particles were examined but shown unsuccessful in our laboratory (De Vos et al. 38, in preparation). Silver particles have the disadvantage (as opposed to e.g. titanium dioxide particles) that application of extreme conditions (pH, added energy, dispersion medium,…) risk to introduce artificial particles (de novo synthesis of particles from Ag ions, breakdown of large particles.
In our laboratory, we further have applied the approach proposed by the reviewer for a series of 10 pristine E174 additives and 10 E174 containing products, including the samples investigated in this study.
The proposed approach does not work to determine extraction efficiency for this type of samples because they are too polydisperse in size.
Very few, relatively large particles (micrometer to millimeter range) contain most of the mass. They are so few that they are hardly visible on number-based distributions.
Comparison of the mass of the nanosized particles, measured by SP-ICP-MS, with the total silver mass, measured by ICP-MS, showed that the mass fraction of the nano-sized particles is in the order of 0.16% - 0.55%. The large majority of the particles that are eluted (by number) contain a very small fraction of the mass.
To address this comment, a clarifying paragraph was introduced in the discussion, after line 408.
“The applied sample preparation is based on the protocol of Jensen et al. [21], a well-established sample preparation protocol for nanotoxicological testing. It is a compromise between very mild and harsch sample preparation, and cannot exclude adsorption of particles to food compounds and to larger Ag particles. The proposed measurement uncertainties associated with the applied sample preparation cannot be extrapolated directly to mild sample preparation, such as elution with water only [3] with possible extensive adsorption, and to harsh sample preparation protocols, that can induce de novo formation of particles or particle breakdown due to the relative physical and chemical instability of silver [28].”
Following reference was introduced:
[28] Potter, P.M., et al., Transformation of silver nanoparticle consumer products during simulated usage and disposal. Environmental Science: Nano, 2019. 6(2): p. 592-598.

Reviewer 3 Report
For publication in Materials, I received the paper Estimation of the uncertainties related to the measurement of the size and quantities of individual silver nanoparticles in confectionery. The article presents research and its results on Ag nanoparticles used as a food additive, in an attempt to validate methods to size and quantify these particles. The topic is worth investigating, the paper is well designed and technically sound. My comments are:
Would missing NPs fraction in ICP-MS compared to TEM analyses not present an obstacle for valid results? FIG 1 (b,c) – are large black regions on TEM images, especially in fig. b), remains of larger agglomerations, is this a support grid, or is this also a particle of interest? Was it included in the size measurements? On FIG c) left centre side I can see at least five particles in size of 100 nm, while on particle size distribution graph there are none. Is there any method to exclude them from measurements, if this is not Ag? And if this is Ag, how come it is not included in the particle size measurements? Next comment refers on previous: any supporting information from EM, that these are only Ag NPs (from “real” samples, this is always a question, especially in confectionery I would expect everything, from TiO2 onwards). The paper is referring in some points on the Supplementary file, not provided with the review The uncertainty increase in complex samples is quite high. Are such results even useful, taking into account the high repeatability variation?Author Response
Reviewer 3
Would missing NPs fraction in ICP-MS compared to TEM analyses not present an obstacle for valid results?
In the part Trueness/recovery (Lines 201-205) it is indicated that “Single particle ICP-MS is considered to be a screening method in nanoparticle analysis because of 1) its assumptions about the shape and composition of the nanoparticles, and 2) its relatively high size detection limits which might result in a substantial amount of nanoparticles not being detected”. In practice spICP-MS results can be complemented with TEM results to better estimate the number-based size distribution.
For clarification, following sentence is added on line 204: “Single particle ICP-MS results can be complemented with TEM results to better estimate the number-based size distribution.”
FIG 1 (b,c) – are large black regions on TEM images, especially in fig. b), remains of larger agglomerations, is this a support grid, or is this also a particle of interest? Was it included in the size measurements?
The described features are micrometer-sized silver flakes of E174. These are not included in the size measurement since their size is much larger than the upper limit of quantification of the EM and spICP-MS analyses. This has a negligible effect on the estimation of the (median) size since the number of such flakes is very low.
For clarification, the following paragraph is introduced in the 2.3. Instrumentation and analysis section after line 186:
“Large micrometer-sized silver flakes of E174 are not included in the size measurement since they are much larger than the upper limit of quantification of the EM and spICP-MS analyses. This has a negligible effect on the estimation of the (median) size since the number of such flakes is low compared to the number of (nano)particles.”
On FIG c) left centre side I can see at least five particles in size of 100 nm, while on particle size distribution graph there are none. Is there any method to exclude them from measurements, if this is not Ag? And if this is Ag, how come it is not included in the particle size measurements?
These particles are included in the size measurement. Because of their low relative frequency they are part of the tail of the log-normal, normalized size distribution (Figure 1C).
Next comment refers on previous: any supporting information from EM, that these are only Ag NPs (from “real” samples, this is always a question, especially in confectionery I would expect everything, from TiO2 onwards).
To address this comment, following sentence was added after line 188: “Silver particles measured by TEM analysis were identified based on their mass-thickness contrast and morphology. The chemical composition of the measured particles was verified using scanning TEM (STEM) combined with energy dispersive X-ray spectrometry (EDX) analysis on representative samples [29].”
The paper is referring in some points on the Supplementary file, not provided with the review
Possibly, something went wrong with the transfer of information. Supplementary files were submitted for review.
The uncertainty increase in complex samples is quite high. Are such results even useful, taking into account the high repeatability variation?
The presented uncertainties are in the line of what is obtained by other authors, as stipulated in line 391.
Following clarification is introduced (after line 391): “Although these uncertainties expressed as percentage seem high, they are only a few nanometer in absolute terms. Results, with uncertainties, are essential for product characterization, in a regulatory context and in risk analysis.”
